

# The effect of yoga and aerobic exercise on children's physical activity in rural India: a randomized controlled trial

Tarun Reddy Katapally[1,2,3,4], Jamin Patel[1,2], Sheriff Tolulope Ibrahim[2,3], Sonal Kasture[4], Anuradha Khadilkar[4] and Jasmin Bhawra[4,5]

[1] Department of Epidemiology and Biostatistics, Schulich School of Medicine and Dentistry, University of Western Ontario, London, Ontario, Canada
[2] DEPtH Lab, Faculty of Health Sciences, University of Western Ontario, London, Ontario, Canada
[3] Children's Health Research Institute, Lawson Health Research Institute, London, Ontario, Canada
[4] Hirabai Cowasji Jehangir Medical Research Institute, Pune, Maharashtra, India
[5] CHANGE Research Lab, School of Occupational and Public Health, Toronto Metropolitan University, Toronto, Ontario, Canada

Corresponding author
Tarun Reddy Katapally,
tarun.katapally@uwo.ca

## ABSTRACT

**Purpose**. This study aimed to assess the impact of yoga and aerobic exercise on moderate-to-vigorous (MVPA) levels of children in rural India.

**Methods**. The study utilized secondary data from a randomized, controlled, open-labelled, single-center, two-site, parallel-group trial. The study was conducted in rural India over a 6-month period between 2018–2019. Children aged 6 to 11 years were randomized into three groups: aerobic exercise (30 minutes, 5 days/week), control (no intervention), and yoga (30 minutes, 5 days/week). MVPA was measured at baseline and at six months using the Quantification of Physical Activity in School Children and Adolescents survey adapted and validated for Indian children. Overall sample and gender-segregated data were analyzed using paired sample t-tests and one-way analysis of variance with *post-hoc* analyses.

**Findings**. In the overall sample ($N = 151$), mean MVPA (minutes/day) increased significantly in both yoga ($n = 50$; $p < 0.001$) and aerobic exercise ($n = 49$; $p < 0.001$) groups from baseline to endline. Among males, mean MVPA increased significantly from baseline to endline in all three groups, including the control group ($n = 23$; $p = 0.005$). Among females, mean MVPA increased only in the yoga group, with baseline to endline change being significant across the three groups ($p = 0.005$), and with the yoga group depicting greater change in comparison to the control group ($p = 0.004$).

**Conclusions**. Our findings suggest that both yoga and aerobic exercise can increase MVPA among rural children, with yoga being particularly beneficial for girls—a significant finding to inform culturally-appropriate active living policies to minimize the current physical activity gender gap in India. These findings can have implications for public health programs and policies not only in India but across other rural areas worldwide, where similar challenges in promoting physical activity among children may exist.

## INTRODUCTION

Physical inactivity is a major contributor to global non-communicable diseases (NCD), (*Katzmarzyk et al., 2022*; *Lee et al., 2012*; *Santos et al., 2023*) with an estimated mortality burden of 3.2 million deaths per year (*Shimizu, 2024*). Physical inactivity is a significant concern in pediatric populations as it has been linked to the prevalence of NCDs in adulthood (*Bull et al., 2020*; *World Health Organization (WHO), 2022*; *Poitras et al., 2016*). Between 2020 and 2030, the cumulative economic cost of India's physical inactivity is projected to be USD 35.4 billion (*World Health Organization (WHO), 2022*). Evidence indicates that physical activity among children and youth in India has consistently decreased since 2016, especially during the coronavirus disease pandemic (*Katapally et al., 2016*; *Bhawra et al., 2018*; *Bhawra et al., 2023*). This decline in physical activity poses a critical challenge for pediatric healthcare providers and policymakers in promoting and facilitating active lifestyles among children (*Lobelo et al., 2020*; *Pellerine et al., 2022*; *Muth et al., 2024*).

Child and youth physical activity is below recommended guidelines (*Katapally et al., 2016*; *Bhawra et al., 2018*; *Bhawra et al., 2023*) despite growing policies and programs (*Bhawra et al., 2023*), which range from school educational curricula and governmental investments in sports infrastructure (*Bhawra et al., 2023*) to comprehensive sports programs (*Khelo India, 2022*), and physical education interventions across India (*Sportz Village Schools, 2024*; *Fit India, 2024*). Culturally-appropriate strategies, which consider the identity and heritage of global south populations need to be considered seriously to reverse this growing physical inactivity trend (*Bhawra et al., 2023*; *Joo & Liu, 2021*; *Horne et al., 2018*; *Leach et al., 2024*). Given its historical and cultural significance, yoga has the potential to serve as a complementary medicine practice to promote physical activity among Indian children in particular (*Bhawra et al., 2023*; *Leach et al., 2024*; *Patel et al., 2024a*). Both breathing (pranayama), and physical techniques (asanas) (*Govindaraj et al., 2016*) have been shown to improve motor function (*Telles et al., 2013*; *Folleto, Pereira & Valentini, 2016*; *Barnett et al., 2024*), muscle strength (*Telles et al., 2013*; *Kasture et al., 2023*), and mental health (*Telles et al., 2013*; *Hart et al., 2022*; *Kauts & Sharma, 2009*) among Indian children and youth.

While the benefits of yoga are well-established, irrespective of the type of yoga practice (*Cramer et al., 2016*), currently there is no concrete evidence linking yoga with moderate to vigorous physical activity (MVPA) among children and youth—a key indicator informing global physical activity guidelines (*US Department of Health Human Services, 2022*; *World Health Organization (WHO), 2020*). Some preliminary correlational evidence is being explored (*Patel et al., 2024a*), however; it is important to establish empirical evidence *via* randomized controlled trials to understand these relationships better. We hypothesize that while the practice of yoga itself can result in the accumulation of more MVPA (*Larson-Meyer, 2016*), yoga practice can inherently influence other active living behaviours

(*Wang et al., 2020*; *Polsgrove, Eggleston & Lockyer, 2016*; *Watts et al., 2018*), which might enhance MVPA. However, it is critical to test this hypothesis in parallel with the effect of aerobic exercise on MVPA, with the obvious hypothesis being that aerobic exercise would increase MVPA (*Bull et al., 2020*; *US Department of Health Human Services, 2022*). It is also important to understand gender variations of the impact of yoga on MVPA to inform healthcare strategies, policies, and programs that promote equitable promotion of MVPA among girls in India—who are generally less physically active than Indian boys (*Bhawra et al., 2018*; *Bhawra et al., 2023*). Thus, as part of a randomized controlled study, our objective is to investigate the influence of yoga and aerobic exercise on MVPA of children in rural India, while exploring gender variations of this effect.

## METHODS

### Study design and setting

This study is a secondary analysis of a randomized, controlled, open-labelled, single-centre, 2-site, three-arm (allocation ratio: 1:1:1) parallel-group trial of yoga, aerobic exercise, and protein supplementation conducted among 232 children (aged 6 to 11 years) (*Kasture et al., 2023*). Several measures were implemented to minimize potential bias arising from the open-label design. The statistician was blinded to group allocation until all analyses were completed, and one investigator was blinded to group identity during the interpretation of de-identified data. Additionally, outcome assessments were conducted using pre-specified protocols to ensure consistency and reduce assessment bias. Children were recruited in two randomly selected government schools in two villages located 60km from Pune, India from July 13th to 25th 2018. Parents provided written informed consent, while their children provided written assent. Ethics approval was obtained from the Hirabai Cowasji Jehangir Medical Research Institute Ethics Review Committee (ECR/352/Inst/MH/2013/RR-16). The study was registered with the Clinical Trials Registry—India (CTRI/2018/07/014815). Figure 1 displays the recruitment, randomization, and group allocation process to determine the final sample size. Additional methodological information, including sample size calculations, has been published by *Kasture et al. (2023)*. Portions of this text were previously published as part of a preprint (*Katapally et al., 2024*).

### Inclusion and exclusion criteria

Male children between the ages of 7 and 11 years and female children between the ages of 6 and 10 years were included, as evidence indicates that female children attain puberty sooner (*McDowell, Brody & Hughes, 2007*; *Cleveland Clinic, 2024*). Additionally, children had to provide MVPA data at baseline and endline to be included in the analyses of this study. Children using medication known to affect bone/muscle health or with chronic illnesses were excluded from the study (*Kasture et al., 2023*). Participant attendance was tracked at each session to assess compliance, and those with a compliance rate of 50% or lower were excluded from the final analysis.
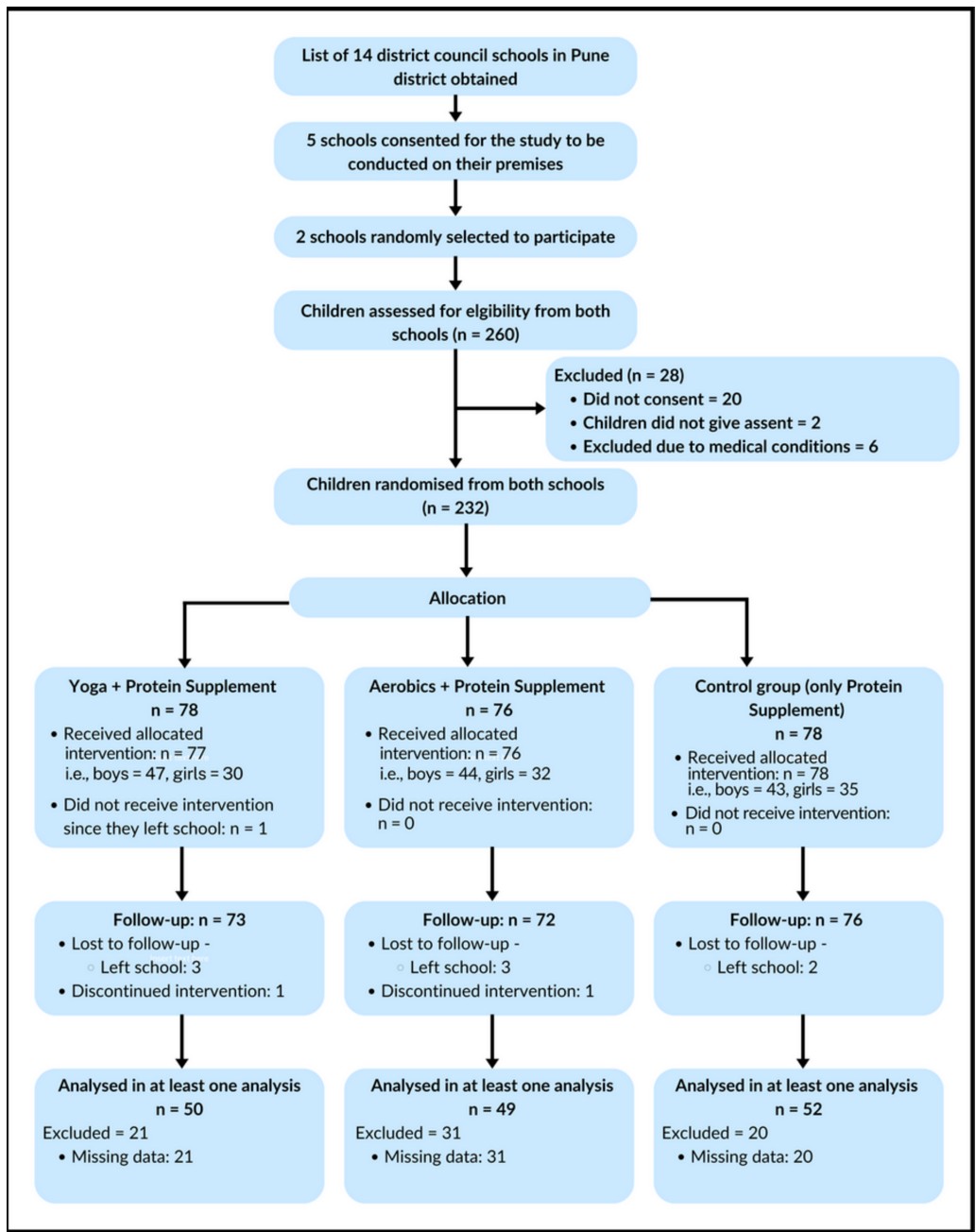

**Figure 1  CONSORT diagram depicting study design and participant flow.**

## Recruitment and randomization

Before enrolment, parents of children ($N = 260$) who met the age criteria were approached. Only 240 parents and their children provided written informed consent and assent, respectively. To achieve randomization, a data manager who was not involved in data collection generated a random sequence *via* a web-based application (*Random.org, 2024*)

to randomize children into three mutually exclusive groups (yoga, aerobic exercise, and control).

## The six-month intervention

All children across the three randomized groups (control, yoga, and aerobic exercise) received a protein supplement (Ladoo) for 6 days a week over six months. The mean nutrient content per serving of Ladoo was 148 kcal energy, seven g of protein, four g of fat, 22 g of carbohydrate, six mg of iron, and 226 mg of calcium. The ingredients for the protein supplement were roasted groundnuts, roasted sesame seeds, fresh dates, skimmed milk powder and powdered cane sugar (*Kasture et al., 2023*). Children in the yoga group engaged in a 30-minute, five-day-a-week yoga exercise (Fig. 2) before school. This exercise was a type of yoga called Yogasanas, which involved engagement in various static poses. The poses required children to use their major joint and muscle groups, including their trunk, back, hip and knee muscles.

Children in the aerobic group engaged in aerobic exercise for 30 min, five days a week (Fig. 2). Daily physical exercise involved a five-minute warm-up, a 20-minute core and lower limb strengthening exercise, and a five-minute cool-down period. In week one, instructors provided hands-on support and visual demonstration to ensure correct posture for children in both groups. Thereafter, visual demonstration and verbal guidance were used to engage with children. Participants in the control group were advised to continue their usual daily routines without any structural physical activity intervention.

## Data collection

Data collection was obtained both at baseline (before randomization) and endline (after 6 months). Age, gender, body mass index (BMI), sunlight exposure, dietary intake data, as well as data on the primary outcome variable, MVPA, were obtained *via* researcher-administered validated surveys in the presence of children's caregivers. The Quantification of Physical Activity in School Children and Adolescents survey adapted and validated for Indian children was used to obtain participants' MVPA data (*Barbosa et al., 2007*; *Khadilkar et al., 2018*). The authors received permission to use this instrument from the copyright holders. The MVPA data was obtained twice, at baseline before randomization, and at the end of the 6 months (*i.e.*, endline). Anthropometry and body composition, including height and weight, were measured using bioelectrical impedance analysis Tanita Body Composition Analyzer (Model BC-420MA).

## Data analysis

All statistical analyses were conducted in R 4.2.1 (*R Core Team, 2021*). Mean and standard deviation were used to describe age, BMI, gender, and minutes of MVPA/day. Paired sample *t*-tests assessed differences in mean MVPA/day within the yoga, aerobic, and control groups from pre- to post-intervention. Mean differences (MD), defined as the difference in mean MVPA/day between pre-and post-intervention, were calculated for the three groups. Analysis of variance with post hoc Tukey tests was used to detect differences in baseline characteristics and MVPA at both pre-and post-intervention, as well as MD

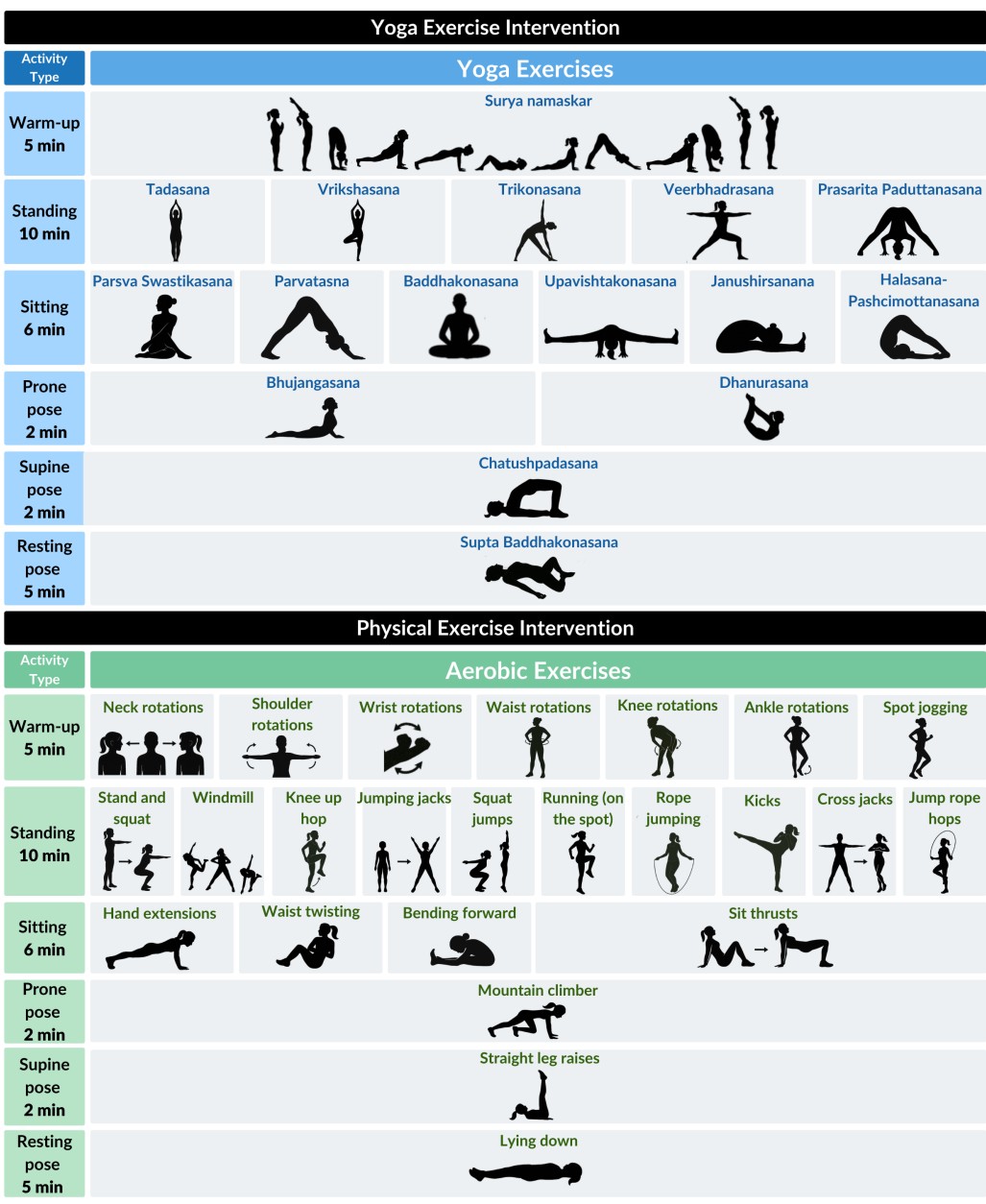

**Figure 2** Daily yoga and aerobic exercise intervention regimen.

between the three groups. Data analyses were segregated by gender (male *vs.* female). Results were deemed statistically significant at $p < 0.05$.

## RESULTS

A total of 151 (N) children were included in the final analyses, with the following distribution across the three groups: yoga ($n = 50$), aerobic exercise ($n = 49$), and control ($n = 52$). The sample comprised of 54.30% males and 45.70% females. Table 1 depicts

no statistically significant differences in the distribution of age and gender between the three groups, both at baseline and endline. However, there was a statistically significant difference in BMI across the three groups at endline ($p = 0.015$). The mean BMI was 14 at baseline and 14.38 at endline, with children in the aerobic exercise group having higher body mass index at both time points.

Table 2 depicts pre- (baseline) and post-intervention (endline) mean MVPA/day differences within and across the three groups (yoga *vs* aerobic *vs* control). These differences are shown in the overall sample, as well as in the gender-segregated samples (males and females). The mean MVPA/day in the overall sample was 54.02 minutes/day and 80.15 minutes/day at baseline and endline, respectively. In the overall sample, mean MVPA/day increased significantly between baseline and endline within both yoga (mean difference (MD) = 35.85, 95% confidence interval (CI) [18.77–52.92]) and aerobic exercise groups (MD = 32.21, 95% CI [18.15–46.27]). There was no significant difference in mean MVPA/day between yoga, aerobic and control groups either at baseline or endline; however, the yoga group depicted higher MD from baseline to endline ($p = 0.057$).

Among males, the mean MVPA/day was 57.53 minutes/day and 98.01 minutes/day at baseline and endline, respectively. Moreover, mean MVPA/day among males increased from baseline to endline among all three groups: yoga (MD = 35.51, 95% CI [11.47–59.56]), aerobic exercise (MD = 45.97, 95% CI [31.64–60.31]), and control (MD = 40.51, 95% CI [13.05–67.96]). There was no significant difference in mean MVPA/day between the three groups at baseline or endline; however, the aerobic group showed higher MD from baseline to endline ($p = 0.779$). Among females, the mean MVPA/day was 49.86 minutes/day and 58.93 minutes/day at baseline and endline, respectively. More importantly, among females, mean MVPA/day increased significantly between baseline and endline only within the yoga group (MD = 36.40, 95% CI [11.52–61.26]), with MD of MVPA from baseline to endline being significant not only across the three groups ($p = 0.005$) but also being higher among the yoga group in comparison to control ($p = 0.004$) and aerobic exercise ($p = 0.328$) groups. Figure 3 shows three sets of boxplots and density plots comparing the distribution of MVPA/day at both baseline and endline across all three groups segregated by overall, male, and female samples. In the overall sample, across all three groups, the distribution of MVPA/day was skewed at baseline, but the skewness was reduced at endline. In both gender-segregated samples, a similar skewness in the distribution of MVPA/day was observed in both yoga and aerobic exercise groups at baseline. Nevertheless, post-intervention (*i.e.,* at endline), the skewness in the distribution of MVPA/day reduced—similar to the overall sample, in both males and females.

## DISCUSSION

This study explored the effect of both yoga and aerobic exercise on children's MVPA. The findings showed that both yoga and aerobic exercise, when consistently incorporated into a randomized controlled trial over a six-month period, led to statistically significant increases in MVPA levels among rural children aged 6–11 years. It is also important to note that children who did not receive either the yoga or aerobic exercise intervention (control group)

**Table 1  Demographic and weight status distribution at baseline and endline.**

| Variables | Baseline | | | | | Endline | | | | |
|---|---|---|---|---|---|---|---|---|---|---|
| | Overall (N = 151) Mean (SD) | Yoga (N = 50) Mean (SD) | Aerobic (N = 49) Mean (SD) | Control (N = 52) Mean (SD) | *p*-value | Overall (N = 151) Mean (SD) | Yoga (N = 50) Mean (SD) | Aerobic (N = 49) Mean (SD) | Control (N = 52) Mean (SD) | *p*-value |
| Age (years) | 9.00 (0.96) | 8.99 (1) | 9.15 (1.05) | 8.88 (0.83) | 0.372 | 9.65 (0.96) | 9.64 (1) | 9.80 (1.05) | 9.53 (0.83) | 0.394 |
| BMI[*] | 14.00 (1.76) | 13.75 (1.91) | 14.51 (1.68) | 13.78 (1.62) | 0.050 | 14.38 (1.91) | 14.09 (1.85) | 15.03 (1.92) | 14.05 (1.82) | 0.015[*] |
| Gender | *n* (%) | *n* (%) | *n* (%) | *n* (%) | *p*-value | *n* (%) | *n* (%) | *n* (%) | *n* (%) | *p*-value |
| Male | 82 (54.30) | 31 (62) | 28 (57.14) | 23 (44.23) | 0.242 | 82 (54.30) | 31 (62) | 28 (57.14) | 23 (44.23) | 0.242 |
| Female | 69 (45.70) | 19 (38) | 21 (42.86) | 29 (55.77) | 0.204 | 69 (45.70) | 19 (38) | 21 (42.86) | 29 (55.77) | 0.204 |

**Notes.**

BMI, body mass index: weight in kilograms/(height in metres)$^2$

*Significant at $p < 0.05$; M, Mean; CI, Confidence Interval; SD, Standard deviation; MD, Mean difference.

Katapally et al. (2025), *PeerJ*, DOI 10.7717/peerj.19604

**Table 2  Mean MVPA within and between groups pre- and post-intervention: overall and gender-segregated samples.**

| Strata | Variable | Yoga (SD) $n = 50$ | Aerobics, (SD) $n = 49$ | Control, (SD) $n = 52$ | ANOVA (p-value) | Aerobic × Yoga (p-value) | Yoga × Control (p-value) | Aerobic × Control (p-value) |
|---|---|---|---|---|---|---|---|---|
| Overall $(N = 151)$ | MVPA (baseline)– $(M = 54.02)$ | 51.11 (50.52) | 46.14 (37.63) | 64.24 (39.27) | 2.425 ($p = 0.092$) | −4.977 ($p = 0.832$) | −13.130 ($p = 0.272$) | −18.107 ($p = 0.089$) |
| | MVPA (endline)– $(M = 80.15)$ | 86.96 (44.42) | 78.35 (36.61) | 75.31 (45) | 1.035 ($P = 0.358$) | −8.613 ($p = 0.570$) | 11.652 ($p = 0.348$) | 3.039 ($p = 0.931$) |
| | *t*-test, (p-value), [95% C.I] | 4.219 ($p < 0.001$)*, [18.77, 52.92] | 4.608 ($p < 0.001$)*, [18.15, 46.27] | 1.369 ($p = 0.177$), [−5.16, 27.28] | | | | |
| | Mean difference (SD) | 35.85 (60.08) | 32.21 (48.94) | 11.06 (58.26) | 2.918 ($p = 0.057$) | −3.636 ($p = 0.944$) | −24.783 ($p = 0.069$) | −21.147 ($p = 0.144$) |
| Males $(N = 82)$ | Variable | Yoga (SD) $n = 31$ | Aerobics, (SD) $n = 28$ | Control, (SD) $n = 23$ | ANOVA (p-value) | Aerobic × Yoga (p-value) | Yoga × Control (p-value) | Aerobic × Control (p-value) |
| | MVPA (baseline)– $(M = 57.53)$ | 62.33 (54.33) | 43.60 (26.44) | 68.01 (47.38) | 2.185 ($p = 0.119$) | −18.730 ($p = 0.246$) | −5.685 ($p = 0.888$) | −24.415 ($p = 0.132$) |
| | MVPA (endline)– $(M = 98.01)$ | 97.84 (49.62) | 89.57 (39.06) | 108.5 (35.92) | 1.249 ($p = 0.292$) | −8.267 ($p = 0.738$) | −10.683 ($p = 0.635$) | −18.950 ($p = 0.260$) |
| | *t*-test (p-value), [95% C.I] | 3.02 ($p = 0.005$)* [11.47, 59.56] | 6.58 ($p < 0.001$)* [31.64, 60.31] | 3.06 ($p = 0.006$)* [13.05, 67.96] | | | | |
| | Mean difference (SD) | 35.51 (65.56) | 45.97 (36.98) | 40.51 (63.49) | 0.25 ($p = 0.779$) | 10.463 ($p = 0.760$) | 4.998 ($p = 0.945$) | −5.465 ($p = 0.938$) |
| Females $(N = 69)$ | Variable | Yoga (SD) $n = 19$ | Aerobics, (SD) $n = 21$ | Control, (SD) $n = 29$ | ANOVA (p-value) | Aerobic × Yoga (p-value) | Yoga × Control (p-value) | Aerobic × Control (p-value) |
| | MVPA (baseline)– $(M = 49.86)$ | 32.82 (38.18) | 49.52 (49.32) | 61.26 (32) | 2.959 ($p = 0.059$) | 16.704 ($p = 0.382$) | −28.437 ($p = 0.046$) | −11.732 ($p = 0.558$) |
| | MVPA (endline)– $(M = 58.93)$ | 69.21 (27.03) | 63.38 (27.29) | 48.97 (32.38) | 3.049 ($P = 0.054$) | −5.830 ($p = 0.807$) | 20.245 ($p = 0.059$) | 14.415 ($p = 0.211$) |
| | *t*-test (p-value), [95% C.I] | 3.07 ($p = 0.007$)* [11.52, 61.26] | 1.11 ($p = 0.281$) | −1.59 ($p = 0.124$) | | | | |
| | Mean difference (SD) | 36.40 (51.60) | 13.86 (57.30) | −12.29 (41.69) | 5.674 ($p = 0.005$)* | −22.533 ($p = 0.328$) | −48.681 ($p = 0.004$) | −26.148 ($p = 0.164$) |

**Notes.**

*Significant at $p < 0.05$; M, Mean; CI, Confidence Interval; SD, Standard deviation; MD, Mean difference.

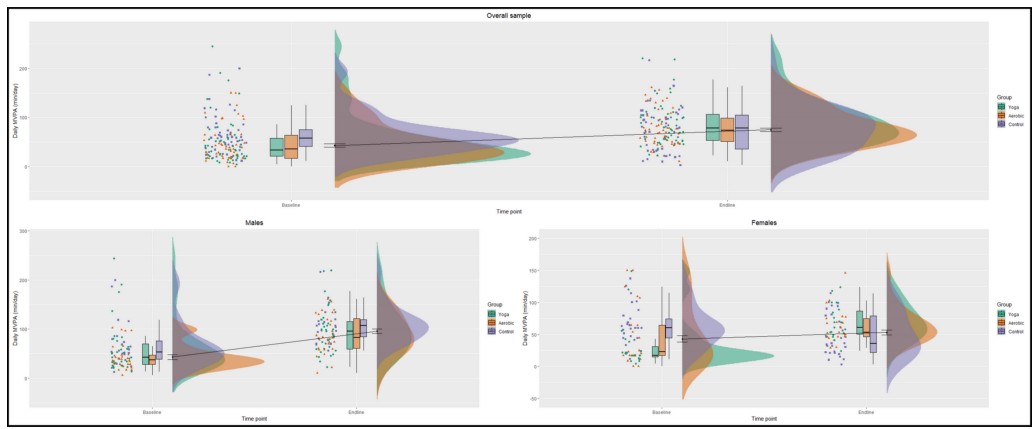

**Figure 3  Changes in MVPA /day distribution from baseline to endline.** Yoga: green, aerobic exercise: orange, control: purple.

did not show a statistically significant increase in their MVPA levels between baseline and endline. However, there were no differences in pre/post-intervention MVPA levels between the yoga and aerobic exercise groups. Nevertheless, more nuanced findings were observed after conducting gender-segregated analyses. Among males, MVPA increased significantly from baseline to endline in all three groups, including the control group. Among females, MVPA increased from baseline to endline only in the yoga group, with the change in mean MVPA from baseline to endline being significantly higher in the yoga group in comparison to the control group. Although the observed increase in MVPA among females following the yoga intervention is a positive finding, the relatively wide confidence interval suggests some uncertainty in the precision of the effect estimate. This may reflect factors such as small sample size, participant heterogeneity, or unmeasured confounders. The benefits of aerobic exercise interventions on MVPA levels among children are well established (*Bull et al., 2020*; *Gordon et al., 2013*; *van der Fels et al., 2020*), and this study adds novel insights to the existing evidence by examining the effect of aerobic exercise on rural Indian children. While there is an emerging body of evidence—particularly the use of randomized control trials—to assess the effects of yoga on physical fitness and mental health (*Telles et al., 2013*; *Hart et al., 2022*; *Varambally et al., 2024*; *Kongkaew et al., 2018*; *Lang et al., 2021*; *Bieber et al., 2021*; *Yadav et al., 2024*), a gap in evidence remains regarding the effect of yoga practice on the overall accumulation of MVPA among children. Thus far, to our knowledge, no study has examined the impact of yoga practice on the MVPA levels of children, who are at a critical stage both in terms of physiological development as well as the formation of active living behaviours.

There is some evidence from cross-sectional studies showing that yoga is associated with higher levels of MVPA in adult populations (*Watts et al., 2018*). Among children, studies have primarily focused on the general health benefits of yoga, such as improvements in mental health (*Telles et al., 2013*; *Hart et al., 2022*; *Kauts & Sharma, 2009*), flexibility

(*Donahoe-Fillmore & Grant, 2019*), and stress reduction (*Kauts & Sharma, 2009*; *James-Palmer et al., 2020*). This study, however, provides novel insights by linking yoga practice directly to increased MVPA levels, a key indicator of health (*US Department of Health Human Services, 2022*) and the primary measure that informs global guidelines on the physical of children and youth (*World Health Organization (WHO), 2020*). More importantly, our findings reinforce the findings of a cross-sectional study conducted among children and youth across 28 cities and villages in India, which found that yoga practice was associated with more minutes of MVPA per day (*Patel et al., 2024a*).

Another critical finding is that yoga can be a culturally-appropriate alternative or supplement to aerobic exercise, particularly among Indian children and youth (*Bhawra et al., 2023*; *Patel et al., 2024a*; *Kasture et al., 2023*). The potential to implement yoga as part of physical activity programming is a significant finding because physical inactivity among children and youth in India has consistently increased since 2016 (*Bhawra et al., 2018*; *Bhawra et al., 2023*), despite growing active living policies and programs (*Bhawra et al., 2023*). It is time to stem this growing physical inactivity trend by incorporating culturally appropriate and accessible practices such as yoga into routine pediatric care, health promotion strategies, and school health interventions (*Bhawra et al., 2023*; *Joo & Liu, 2021*; *Horne et al., 2018*; *Kasture et al., 2023*).

Yoga, as an alternative to aerobic exercise, might be particularly relevant to rural areas, where resources and physical activity infrastructure in schools may be limited (*Pfledderer et al., 2021*). As yoga is not only culturally relevant but also cost-effective and easy to implement (*Kasture et al., 2023*), there is potential to incorporate its practice into public health strategies (*World Health Organization (WHO), 2018*). Moreover, promoting yoga leverages existing cultural knowledge and practices (*Bhawra et al., 2023*), potentially enhancing community engagement (*Bhawra et al., 2023*; *World Health Organization (WHO), 2018*) and support for physical activity programs (*Patel et al., 2024a*; *Mohanty, Epari & Yasobant, 2020*). As an ancient Indian practice, the incorporation of culturally-appropriate exercises is also relevant in the context of decolonizing physical activity (*Knuth et al., 2024*; *Pang, Balram & Knijnik, 2023*; *Ganneri, 2012*; *Alter, 2000*; *Chakraborty, 2007*), which requires a holistic approach that integrates health and education systems (*Katapally, 2020*). Decolonizing active living entails providing children with diverse physical activity options which recognize their local heritage and culture—which vary widely across the Indian subcontinent *i.e.,* can go beyond just yoga to include dance, and martial arts, among other practices (*Knuth et al., 2024*; *Ganneri, 2012*; *Aubert et al., 2022*). Nonetheless, as our findings do not indicate a difference between aerobic exercise and yoga in improving overall MVPA among children, it is important to note that we do not propose yoga as a replacement for aerobic exercise—which has immense benefits to children's health (*Bull et al., 2020*; *Telles et al., 2013*; *van der Fels et al., 2020*)—rather it is a practice which warrants consideration for physical activity programs in the Indian context.

The findings also indicate gender-specific effects, with MVPA levels increasing from baseline to endline in both the yoga and aerobic exercise groups among males, while MVPA levels increasing only in the yoga group among females. Perhaps most importantly, this change in MVPA from baseline to endline was significant but not in the control group

- suggesting that yoga may be particularly effective for increasing physical activity levels among female children. This finding has particular implications in mitigating and/or minimizing barriers to physical activity accumulation commonly experienced among girls due to cultural norms and societal expectations, particularly in rural India (*Bhawra et al., 2018*; *Bhawra et al., 2023*) These findings underscore the importance of considering gender when designing active living policies and healthcare strategies (*Camacho-Miñano, LaVoi & Barr-Anderson, 2011*), as gender-specific strategies are necessary to ensure active living equity (*Bhawra et al., 2023*; *Bourke et al., 2023*; *Kretschmer et al., 2023*).

Interestingly, when the distribution of MVPA was explored across the three samples (overall, males, and females), we found that the baseline distribution of MVPA levels was skewed across all three groups, with most children engaging in low levels of MVPA. However, in the yoga and aerobic groups, the MVPA levels not only increased, but the distribution became more normally distributed at the six-month follow-up. This indicates that children who were not engaging in sufficient physical activity at baseline not only showed an increase in their MVPA at endline, but also that the MVPA levels were more evenly distributed (*i.e.,* similar). This observation has implications for prompting equity in physical activity to ensure that all children, regardless of their starting point, have the opportunity to improve their health outcomes (*World Health Organization (WHO), 2018*; *Owen et al., 2022*; *Gautam et al., 2023*).

This study found that both yoga and aerobic exercise interventions increased MVPA levels among rural Indian children. Apart from yoga practice and aerobic exercise causing exertion that increases MVPA accumulation, yoga or aerobic exercise also promotes overall physical fitness (*Telles et al., 2013*; *Kasture et al., 2023*) and health (*Poitras et al., 2016*; *US Department of Health Human Services, 2022*; *World Health Organization (WHO), 2018*; *Janssen & LeBlanc, 2010*) which naturally leads to an increase in physical activity levels (*Telles et al., 2013*; *Collins & Staples, 2017*). Moreover, yoga has been shown to improve muscle function (*Telles et al., 2013*; *Kasture et al., 2023*), motor function (*Telles et al., 2013*; *Folleto, Pereira & Valentini, 2016*; *Barnett et al., 2024*), and cardiorespiratory fitness (*Telles et al., 2013*) which can enable children to participate in a broad range of physical activities with greater ease (*Gäbler et al., 2018*).

While both yoga and aerobic exercise might contribute to higher MVPA levels, yoga might be especially pertinent in India not only because it is culturally relevant, but also because it can be practiced in low-resource settings (*Kasture et al., 2023*; *Mohanty, Epari & Yasobant, 2020*; *Frank & Larimore, 2015*; *Dhungana et al., 2021*). This makes yoga accessible to a wider population, particularly in rural regions where active living infrastructure is limited (*Pfledderer et al., 2021*; *Owen et al., 2022*; *Nigg et al., 2022*; *Müller et al., 2024*). Additionally, yoga can be practiced indoors in all weather conditions, an often overlooked component in promoting active living to counter air pollution, regardless of cultural, built environment, gender, and safety barriers (*Katapally et al., 2016*; *Bhawra et al., 2018*; *Bhawra et al., 2023*), which influence MVPA levels among children and youth (*Patel et al., 2024b*; *Katapally, Rainham & Muhajarine, 2015*; *Mitchell, Clark & Gilliland, 2016*).

## STRENGTHS AND LIMITATIONS

The randomized controlled design enables a robust comparison between intervention groups by minimizing potential biases. The inclusion of a culturally appropriate intervention such as yoga, provides novelty and relevance, particularly in developing relevant active living policies for children and youth in India. Moreover, conducting the study in rural Indian schools helps address another gap in evidence (*Katapally et al., 2016*; *Bhawra et al., 2018*; *Bhawra et al., 2023*). Furthermore, the gender-stratified analyses add another layer of strength, providing insights into how tailored active living strategies and practices can be developed to address the gender gap in active living (*Katapally et al., 2016*; *Bhawra et al., 2018*; *Bhawra et al., 2023*). A limitation of this study is the reliance on self-reported MVPA data, which can be subject to recall bias and inaccuracies, potentially affecting the reliability of the measurements.

Additionally, the study was conducted in only two district schools, which may limit the generalizability of the findings to other regions or populations. Our targeted approach allowed for a clearer interpretation of the primary outcome. However, the exclusion of sunlight exposure and dietary intake data may limit the ability to fully account for potential confounding or moderating factors that could influence physical activity levels. These variables should be examined in future studies to provide a more comprehensive understanding of the determinants of MVPA and to explore potential interactions with yoga intervention. Finally, while the six-month follow-up period provides valuable insights into the medium-term effects, it does not capture the long-term sustainability of the interventions. Future studies should be conducted longitudinally over a longer period and with larger, more controlled samples across the diverse geographic regions in India to capture the consistency, magnitude, and broader impact of yoga and aerobic exercise on the MVPA of children. Incorporating objective measures such as an accelerometer will further strengthen the validity and precision of MVPA measurement, thereby reducing recall or reporting bias. Finally, replication studies across other regions of India and the Global South should be carried out to assess the generalizability and long-term sustainability of the findings of this study.

## CONCLUSIONS

To our knowledge, this is the first study to examine the impact of yoga and aerobic exercise on MVPA levels of children. Unlike prior studies centered on structured sports or Western models, our study highlights the value of culturally grounded interventions in a low-income context and offers new insights into gender-sensitive approaches to physical activity. This study demonstrates that both yoga and aerobic exercise can significantly increase MVPA levels among rural Indian children, highlighting their effectiveness in promoting physical activity in resource-limited settings. The findings emphasize the value of culturally-appropriate interventions like yoga, particularly to minimize the gender gap, and enable equity in active living. The findings of this study have important practical implications for integrating low-cost, scalable physical activity interventions into school curricula and community health programs. Policymakers, educators, and public health

practitioners in India and other countries in the Global South could adopt yoga-based interventions as a culturally relevant strategy to promote inclusive, sustainable, and equitable approaches to physical activity. Such initiatives may support broader public health efforts aimed at improving child health outcomes and reducing the burden of physical inactivity-related diseases.

## ACKNOWLEDGEMENTS

We acknowledge Hirabai Cowasji Jehangir Medical Research Institute for their support in obtaining this valuable data, and Heya Desai for her contributions to visualization and collation of this manuscript.

### Funding
This work was funded by the Canada Research Chairs program, which supports Dr. Tarun Katapally's research program. The funders had no role in study design, data collection and analysis, decision to publish, or preparation of the manuscript.

### Grant Disclosures
The following grant information was disclosed by the authors:
Canada Research Chairs program.

### Competing Interests
The authors declare there are no competing interests.

### Author Contributions

- Tarun Reddy Katapally conceived and designed the experiments, performed the experiments, authored or reviewed drafts of the article, and approved the final draft.
- Jamin Patel analyzed the data, prepared figures and/or tables, authored or reviewed drafts of the article, and approved the final draft.
- Sheriff Tolulope Ibrahim analyzed the data, prepared figures and/or tables, authored or reviewed drafts of the article, and approved the final draft.
- Sonal Kasture conceived and designed the experiments, performed the experiments, authored or reviewed drafts of the article, and approved the final draft.
- Anuradha Khadilkar conceived and designed the experiments, performed the experiments, authored or reviewed drafts of the article, and approved the final draft.
- Jasmin Bhawra conceived and designed the experiments, performed the experiments, authored or reviewed drafts of the article, and approved the final draft.

### Human Ethics
The following information was supplied relating to ethical approvals (i.e., approving body and any reference numbers):

Ethics approval was obtained from Hirabai Cowasji Jehangir Medical Research Institute Ethics Review Committee (ECR/352/Inst/MH/2013/RR-16).

## Clinical Trial Ethics

The following information was supplied relating to ethical approvals (i.e., approving body and any reference numbers):

Ethics approval was obtained from the Hirabai Cowasji Jehangir Medical Research Institute Ethics Review Committee (ECR/352/Inst/MH/2013/RR-16).

## Data Availability

The anonymized raw data required to reproduce the findings are available at Figshare: Ibrahim, Sheriff; Patel, Jamin; Katapally, Tarun (2024). Yoga vs. aerobic exercise in increasing MVPA among children: secondary analysis of a randomized controlled trial in rural India. figshare. Dataset. https://doi.org/10.6084/m9.figshare.25970053.v1.

## Clinical Trial Registration

The following information was supplied regarding Clinical Trial registration:

The study was registered with the Clinical Trials Registry - India (CTRI/2018/07/014815).

## Supplemental Information

Supplemental information for this article can be found online at http://dx.doi.org/10.7717/peerj.19604#supplemental-information.

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
