# Peer review of "The effect of yoga and aerobic exercise on children’s physical activity in rural India: a randomized controlled trial"

_PeerJ, doi:10.7717/peerj.19604_

## Round 0.1 · original submission · Minor Revisions

Two reviewers gave lots of substantial comments/suggestions, either on methodological insufficient description or on something that needed to be fully discussed. Please refer to comments/suggestions and revise accordingly. I will make my decision when I receive satisfactory recommendations from two reviewers.

**Language Note:** PeerJ staff have identified that the English language needs to be improved. When you prepare your next revision, please either (i) have a colleague who is proficient in English and familiar with the subject matter review your manuscript, or (ii) contact a professional editing service to review your manuscript. PeerJ can provide language editing services - you can contact us at [email protected] for pricing (be sure to provide your manuscript number and title). – PeerJ Staff

·

Basic reporting

-

Experimental design

1. Missing Information on Protein Supplement Dosage
The manuscript mentions that all participants across the three groups received a protein supplement for six days a week over six months. While the ingredients are listed, the manuscript does not specify the amount (e.g., grams, kilocalories, or protein content) provided per serving.
This omission limits the reproducibility of the study and raises the possibility that the supplement may have contributed to differences in MVPA outcomes across groups.

Recommendation:
Please specify the exact amount and nutritional content of the protein supplement administered to participants, or explain the rationale for its omission if this information was not recorded.

2. Lack of Reporting on Sunlight Exposure and Dietary Intake
In the Methods section, the authors mention collecting data on sunlight exposure and dietary intake. However, these variables are not reported or discussed in the Results or Discussion sections.
As these factors may influence physical activity levels or serve as confounders, their absence from the analysis or interpretation should be clarified.

Recommendation:
Please clarify whether the sunlight exposure and dietary intake data were analyzed. If they were excluded, kindly provide justification or indicate if they will be addressed in a future publication or supplementary file.

Validity of the findings

1. Wide Confidence Interval in Female Yoga Group
The manuscript reports statistically significant improvements in MVPA following the yoga intervention, with a reported mean difference (MD) of 36.40 minutes/day (95% CI: 11.52 to 61.26).
While the result is promising, the relatively wide confidence interval suggests considerable uncertainty regarding the precision of the estimated effect size.

Recommendation:
I recommend that the authors acknowledge this limitation in the Discussion section and consider potential contributing factors, such as sample size, population heterogeneity, or unmeasured confounders. Future studies with larger and more controlled samples may help provide more precise estimates of the intervention effect.

·

Basic reporting

The introduction is well-structured, professionally written, and provides a strong foundation for the study. It effectively highlights the significance of physical inactivity, particularly in pediatric populations, and supports the discussion with well-cited literature. The exploration of yoga as a culturally relevant intervention is insightful, and the hypotheses are clearly stated, linking yoga and aerobic exercise to MVPA while addressing gender variations. The writing is clear, logical, and aligns well with academic standards. With minor refinements in clarity and flow, this introduction sets a solid stage for the study.

Experimental design

The study is well-structured, presenting a clear research question, a rigorous three-arm RCT design, and adherence to high ethical standards. Ethics approval and trial registration enhance its credibility, while well-defined inclusion criteria, detailed intervention protocols, and a transparent randomization process support replicability. To further strengthen the study, consider expanding on control group activities beyond protein supplementation, incorporating adherence tracking through attendance records, specifying MVPA measurement tools and frequency, and discussing strategies to mitigate bias in the open-label design. These refinements would improve clarity and reinforce the study’s methodological rigor and impact.

Validity of the findings

The study presents robust and statistically sound findings, with all underlying data meticulously controlled and transparently provided. The conclusions are clearly articulated, directly aligned with the research question, and well-supported by the results. To further enhance validity, it is recommended to explicitly highlight the impact and novelty of the findings within the broader academic discourse. Additionally, discussing the rationale for potential replication studies would reinforce the study’s contribution to the field. Addressing any limitations in data interpretation, ensuring transparency in statistical methodologies, and detailing potential sources of bias will further strengthen the credibility and applicability of the research.

Additional comments

Here are some general comments that are not covered by the three specific areas:

Clarity and Readability – The study is well-structured and effectively conveys complex concepts. However, ensuring that technical terms are clearly defined and consistently used throughout will enhance accessibility for a broader audience.

Practical Implications – While the study presents strong theoretical contributions, elaborating on real-world applications and implications could further demonstrate the research's impact beyond academia.

Future Research Directions – Providing more explicit recommendations for future research could help guide subsequent studies and emphasize the study’s role in shaping ongoing discussions in the field.

---

## Round 0.2 · accepted · Accept

Two reviewers agree with most of your revision. However, one reviewer suggested that you need to check your English writing. I agree with this suggestion. Please ask an English expert to proofread your manuscript.


·

Basic reporting

No comment

Experimental design

No comment

Validity of the findings

No comment

Additional comments

No comment

·

Basic reporting

The involvement of a subject-matter expert proficient in English has likely contributed to the improved readability and coherence. The introduction remains strong and now reads more smoothly. No further revisions are needed on this point.

Experimental design

The inclusion of the nutritional content per serving in the Methods section (Lines 135–137) improves the clarity and reproducibility of the study. This added detail also allows for a better understanding of the potential influence of the supplement on MVPA outcomes. No further action is needed on this point.

I appreciate the authors’ decision to narrow the scope of the current analysis to focus on the primary outcome of MVPA. Including a note in the Strengths and Limitations section (Lines 309–314) appropriately acknowledges the potential influence of unreported factors such as sunlight exposure and dietary intake. This addition improves the transparency of the study's limitations and provides helpful direction for future research. No further revisions are needed on this point.

Validity of the findings

The added discussion (Lines 219–222) appropriately acknowledges the uncertainty introduced by the wide confidence interval and identifies plausible contributing factors such as sample size and participant heterogeneity. The mention of the need for larger, more geographically diverse studies (Lines 315–319) further strengthens the manuscript by contextualizing the findings and outlining future research directions. This revision sufficiently addresses my comment. No further changes are necessary on this point.

Additional comments

The manuscript has clearly benefited from careful proofreading, ensuring that technical terms are defined and used consistently, which enhances clarity and readability for a broader audience. The addition of practical implications in the Conclusion effectively connects the study’s findings to real-world applications, especially in the context of public health and education policy in the Global South. Furthermore, the expanded discussion on future research directions provides valuable guidance for subsequent studies, emphasizing the need for longitudinal designs, objective measurement tools, and broader geographic representation. These revisions strengthen the overall impact and relevance of the manuscript.